# Perceived Environmental Attributes: Their Impact on Older Adults’ Mental Health in Malaysia

**DOI:** 10.3390/ijerph19063595

**Published:** 2022-03-18

**Authors:** Teck Hong Tan

**Affiliations:** School of Economics and Management, Xiamen University Malaysia, Sepang 43900, Selangor, Malaysia; waltertan@xmu.edu.my

**Keywords:** health-related quality of life, environmental attributes, urban, Malaysia

## Abstract

In Malaysia, the population of older adults will increase in the coming years. In this context, there is a requirement to build an age-friendly environment to enable the elderly to age healthily. Many studies have shown that a built environment that allows older adults to age in place improves their mental health. However, person-environment analysis that considers mental well-being has remained rare for older adults living in Malaysia. This study examines the relationship between Malaysian seniors’ perceptions of their surroundings at home and in the neighborhood and their mental health. Using stratified sampling, 510 seniors aged 60 and over were interviewed. The results showed that accessibility (*p*-value 0.033, 95% CI for coefficients 0.006, 0.146), environmental qualities (0.015, 0.014, 0.129) and neighborhood problems (0.000, −0.299, −0.146) were significant determinants of elderly people’s mental health. With respect to respondents’ socio-demographic characteristics, female elderly (0.000, 0.616, 0.782), older adults with an elementary education (0.000, 0.263, 0.685) or a college degree (0.026, 0.019, 0.294), being married (0.005, 0.047, 0.259), the ability to drive (0.000, 0.993, 1.315), the number of dependents in the family (0.003, −0.060, −0.012), and homeownership (0.000, −0.602, −0.271) were significantly related to mental well-being.

## 1. Introduction

Urbanization is rapidly increasing in countries around the world. By 2050, the rate is expected to increase to about 70%. This new urban phenomenon could pose a major challenge for older people in promoting aging in place in later life. To address the emerging global challenges related to the aging population, creating an age-friendly environment has become an important goal for countries to pursue as the built environment, which includes age-friendly homes and neighborhoods, has a significant influence in terms of a person’s health at different stages of life. Malaysia is no exception in the problem of aging. In Malaysia, an elderly citizen is someone who is 60 years old or older. Adults aged 60 and above comprised 10.3 percent of the population in 2019 [1]. It is predicted that the proportion of people over the age of 60 will rise to 14% by 2044 and 20 percent by 2056 [2].

The World Health Organization has raised awareness of the importance of a high-quality built environment for seniors, and the Age-Friendly Cities (AFC) concept aims to improve senior adults’ functioning and well-being [3]. According to the Annual Global Retirement Index (GRI), Malaysia is the 15th best place in the world to retire in 2022 [4]. The GRI is a composite retirement security score developed by International Living based on ten evaluation metrics, including housing, cost of living, health care, and climate. Although Malaysia has a reasonably adequate health care system for its citizens, rising health care costs present a burden for the elderly. As reported by the Ministry of Health Malaysia [5], health expenditure per capita increased by 450 percent in nominal terms from RM 393 in 1997 to RM 1790 in 2017. To reduce the health cost burden of the elderly, improvements to the built environment may lower the elderly’s health risk. Older people may suffer from anxiety, disability, and depression due to an unsuitable home and neighborhood environment. Tan and Lee [6] found that the current living environment of Malaysian seniors is not suitable for their needs and requirements in old age. In this regard, there is a need to determine the best environment for senior adults to age in place by examining how environmental characteristics affect the health of Malaysian senior adults.

### 1.1. Home and Neighborhood Environment

Many recent studies have considered both the direct and indirect effects of the attributes of the human-made environment on older adults’ health [7,8,9,10]. Evidence from gerontological studies suggests that mental well-being is affected by home and neighborhood attributes as well as by individual socio-demographic characteristics [11,12,13,14,15]. This evidence forms the basis for the conceptual framework of this study.

The Malaysian government presented the Building Design Guidelines for Malaysian Elderly at the 12th ASEAN-Japan Senior Officials Meeting to enable the design and building of appropriate housing developments and support facilities for Malaysia’s older adults [16]. According to the proposed guidelines, the internal features of senior housing estates should include age-friendly features. Individuals generally spend most of their time indoors. Since most of the time is spent indoors, the characteristics of the home, such as the level of illumination and the structural design of the home, are important to the health of the individual. For example, illuminance is one of the features that needs to be considered when designing the home environment. Adequate lighting of the environment could provide a safe environment for older adults to move around more, which in turn would improve their health outcomes. Kunduraci [17] highlighted the importance of interior lighting design and visual comfort in preventing falls in older adults. It is often suggested that poor lighting can cause accidents and injuries in seniors. In addition, the level of lighting, especially the amount of daylight, could influence an individual’s mental health. It has been found that people who are chronically exposed to less daylight suffer from sadness, fatigue, and clinical depression [7].

The design and structure of an elderly-friendly home are critical in promoting elderly health. Therefore, the elderly-friendly home should be designed according to the principle of aging in place. An accessible entrance, larger doorways and rooms, grab rails, non-slip flooring, appropriate height of electrical switches and a senior-friendly bathroom are all features of the elderly-friendly home. A poorly structured and designed home, for example, without slip-resistant flooring in the bathroom, puts seniors’ safety at risk by causing them physical harm if they fall. Nonfatal bathroom accidents are a common reason for broken bones in older adults [18]. A safe bathroom for the elderly could significantly reduce the likelihood of anger-hostility, tension-anxiety, and depression-dejection [19].

In addition to the home environment, the neighborhood environment of the settlement influences wellbeing in later life [8,15,20,21]. The attributes of the neighborhood environment are classified into two dimensions: physical and social attributes. Physical attributes typically include access to local services or amenities, environmental incidents, such as vandalism, graffiti, litter, poor air quality and traffic, and the availability of a well-connected pedestrian network, while social aspects include feeling safe walking around in the neighborhood and social interaction among older adults in an attractive environment [22,23,24,25,26]. In this regard, this study examines key neighborhood attributes, such as accessibility, safety, environmental quality, neighborhood problems, and pedestrian connectivity and their effects on older adults’ mental health.

There is much evidence to suggest that an accessible and walkable neighborhood can promote older adults’ overall health [27,28,29,30]. Accessibility features include easy access to amenities and proximity to amenities, such as a local store, garden, and health services. It is often assumed that older adults prefer to stay in a neighborhood where they can walk to all major amenities and facilities. As shown in the findings of Fogal et al. [31], older adults from an accessible and walkable neighborhood have a lower risk of functional disability. In addition, such a neighborhood could decrease the risk of developing symptoms of depression [32]. Berke et al. [11] also discovered a noteworthy inverse relationship between depressive symptoms and walkability at the neighborhood level among men as opposed to women.

A pedestrian environment should be encouraged through the provision of sidewalks and alleys to avoid declines in terms of health-related quality of life as people age. A specialized network of pedestrian access, dedicated bicycle linkage, and well-organized public transportation should be incorporated into neighborhood permeability. A well-connected pedestrian network can help reduce car dependence, which should enhance older adults’ overall health status [33,34]. As shown by Parra et al. [9], the overall health status of the elderly was significantly affected by a reduction in traffic in the neighborhood. Empirical studies also show that connectivity in the built environment can significantly affect the quality of life. A reduction in traffic volume could increase residents’ well-being.

An absence of crime in the neighborhood is essential to increase the standard of living for residents, as increasing crime rates can deteriorate the quality of the built environment and reduce neighborhood satisfaction [35]. It is widely assumed that fostering a sense of community belonging is possible in a safe and secure neighborhood, as such a neighborhood creates a safe environment in which residents can participate in social activities. There is a substantial amount of evidence to suggest that neighborhood safety may improve the overall health of older adults [15,29,34]. Fear of crime has been identified as one of the determinants of poor mental and physical health in urban environments [36].

It is widely recognized that public and open spaces within or surrounding residential neighborhoods assist seniors with activities of daily living. Residential neighborhoods are often social communities with a high degree of personal interaction among residents. When residents interact with one another, open spaces play an important role in communication. Mutual support and relationships with family, friends, and neighbors are regarded as key social capital that influence the quality of life, which, in turn, affects senior health [37,38]. Several previous studies have found that good environmental qualities are related to aging adults’ health. Mass et al. [39] found tree-lined streets and parks were associated with higher five-year survival rates in the elderly. In terms of mental health, living in more environmentally friendly neighborhoods, according to Wang et al. [40], may lead to a lower risk of depression.

Neighborhood problems are one of the most crucial aspects when studying the health status of older adults. There are several types of neighborhood problem, including traffic-related noise and air pollution, unclean surroundings, and neighborhoods with vacant and abandoned buildings. Studies by Yen et al. [41] have shown that neighborhood problems can increase the risk of functional decline and limit mobility. For example, air quality is a significant predictor of general health, anxiety, and depression [42]. It is plausible that good air quality may lead to better physical health. Disabilities in daily living activities (ADLs) and health deficits are more likely to exist in those who live in an air-polluted environment [43]. Similarly, noise pollution and the health of the elderly were found to have a significant relationship [9]. Residents from neighborhoods with a large number of vacant and abandoned buildings suffer from health crises, as these structures pose a significant problem to community safety [44].

### 1.2. Socio-Demographic Characteristics

The health of the elderly is thought to be influenced by socio-demographic factors. To reduce selection bias in the estimations, demographic characteristics, such as gender, race, age, marital status, educational background, and household size were controlled for. For example, ‘old-old’ seniors are more likely to have health problems and chronic conditions than ‘young-old’ seniors, who have relatively high levels of health. [45]. Elderly people who are female [46,47], have a lower level of education [46,48,49], are retired [50], who are tenants [50,51], and those who are widowed [48] have a higher prevalence of health risk. Additionally, the ability to drive, as a form of mobility, is beneficial to an individual’s health and well-being [8].

In light of the previous discussion, this paper, therefore, examines the extent to which built-environmental features, such as housing quality, accessibility, pedestrian connectivity, environmental quality, neighborhood problems, and neighborhood safety affect the mental health status of Malaysian older adults.

## 2. Materials and Methods

### 2.1. Respondents and Study Area

Older adults (those aged 60 and above) from Bandar Sunway (Sunway City in English) were surveyed in this study. This township is located in the Malaysian state of Selangor (see Figure 1). In comparison to other townships in Malaysia, this township was chosen as a suitable case study because the township planner addressed the issue of aging in place at two different levels in the municipality, and it is regarded as one of the best neighborhoods for living in Selangor [52]. At the residential level, senior-friendly properties have been built to accommodate the growing elderly population. At the community level, this township is an integrated township with good access to amenities, such as a shopping mall, a hotel, a recreation area, an education center, a health center and a public transport network, that can support aging in place.

The study’s respondents were men and women aged 60 and older who had lived in Bandar Sunway for at least two years because the impact of the perceived built environment determinants on mental well-being in later life would take at least two years to detect [11]. Three districts of Sunway City, namely PJS 7, PJS 9 and PJS 11, were selected as target areas because they are primarily residential areas where the majority of residents live (see Figure 2). The researchers contacted representatives from the various housing projects’ residents’ associations in each neighborhood. Stratified sampling was used to interview respondents from five housing projects identified in PJS 9 and PJS 11, respectively, and three housing projects identified in PJS 7. In total, 510 valid responses were finally used as these responses were checked for outliers and missing values. This represented 38% of the questionnaires distributed.

### 2.2. Variables Used

To study the effects of human-made environment characteristics on Malaysian older adults’ mental health, the questionnaire sought information about respondents’ opinions and experiences of their homes and neighborhood and self-reported mental health, as well as socio-demographic characteristics. In this study, the perceived home environment was measured to assess how well respondents’ needs were met by their current housing situations (i.e., brightness inside the unit, bathroom, outdoor housing conditions) [6,55]. Following previous studies, respondents were asked how they perceived accessibility [56], pedestrian connectivity [56], safety [56], environmental quality [8], and problems [15,57] in the neighborhood. The state of mental health among the elderly in this study was determined by how the respondents perceived their general mental health. Using Radloff’s Center for Epidemiologic Studies Depression Scale (CES-D) items [58], self-reported mental health responses were classified as rare, sometimes, often, and most of the time. The CES-D is thought to be a reliable predictor of depressive symptoms in the elderly [59].

Respondents were asked to provide information about their socio-demographic characteristics, such as gender, age, racial and educational background, employment and marital status, household size, and driving ability. Respondents were also asked whether their home was owned or rented.

The validity and reliability of the instrument used were assessed using confirmatory factor analysis. All construct items met the literature-based criterion, as the values of the loading of the items were greater than 0.5, the average variance extracted values (AVE) were larger than 0.50, the composite reliability values (CR) were larger than 0.70, and the HTMT values for conceptually similar constructs were less than 0.90 (Table 1).

### 2.3. Statistical Analysis

To use a regression model to assess the influence of perceived home and neighborhood environment on older adults’ mental health, after controlling for respondents’ socio-demographic characteristics, the constructs of home and neighborhood characteristics were transformed into composite indices, and the construction of these indices was the average score of the survey items for each construct.

The regression equation can be represented as follows:*MH* = *a* + *b*_1_*H_i_* + *b*_2_*A_i_* + *b*_3_*PC_i_* + *b*_4_*S_i_* + *b*_5_*EQ_i_* + *b*_6_*NP_i_* + *b*_7_*D_i_*
(1)
where*MH* = Self-reported mental health status;*H* = Home environment;*A* = Accessibility;*PC* = Pedestrian connectivity;*S* = Neighborhood safety;*EQ* = Environmental qualities;*NP* = Neighborhood problems;*D* = Demographic characteristics of respondents*a* = constant; and*b*_1_, *b*_2_, …, *b*_7_ = coefficients to be computed

## 3. Results and Discussion

In this survey, the majority of respondents were females between the ages of 60 and 70, having less than a college degree, non-retired, homeowners, married, and driving a car. Malaysia is a multiracial nation. In terms of race, Chinese respondents accounted for 54.9 percent (280) of the total, while Malay and Indian respondents accounted for 33.9 percent (173) and 9.4 percent (48), respectively. Other race respondents made up 1.8 percent (9) of the total. The number of dependents influences the size of the household. An average of 4.32 family members were reported in this study (see Table 2).

Table 3 shows the estimation of the regression model for each of the home and neighborhood environment factors and demographic descriptors on older adults’ mental health status. Taking sociodemographic characteristics into account, the estimates from the model revealed that perceived environmental qualities, as characterized by attractive natural landmarks and buildings in the surrounding area, were significantly associated with mental health (*p*-value 0.015, 95% CI 0.014, 0.129). In other words, older people who lived in locations with good environmental qualities were more likely to have good mental health status. Previous studies have associated access to natural landscapes with improved psychological health. Wang et al. [40] confirmed that living in more green-space-rich neighborhoods can reduce the likelihood of depression. This is because the characteristics of outdoor communal spaces, such as attractive natural sights, may play a part in fostering a stronger sense of belonging to the community through spontaneous social support and social ties to neighbors. This, in turn, promotes health-related well-being in later life [9,38,60,61,62,63].

Perceived neighborhood safety was found to be significantly related to seniors’ mental health status, implying that respondents were concerned about a sense of safety and security when walking around the neighborhood. There is a need to create a safe environment for the elderly to design an age-friendly living environment. Research evidence shows that a low crime rate in a neighborhood can have a positive effect on the elderly’s health-related well-being [9]. In addition, older adults are more likely to take part in regular interpersonal contact if they live in a safer neighborhood [34] because they would choose to go out and meet more neighbors or friends [29]. The study by Balfour and Kaplan [15] also showed that low levels of environmental, physical, and psychological stress are connected with safe environments, all of which have implications for health-related quality of life. A safe neighborhood environment is also beneficial to older adults’ autonomy and sense of direction [64], as well as being low in risk of injury or death [65].

Consistent with previous findings by James et al. [32], Goto et al. [19], and Yu et al. [29], after controlling for demographic factors, access to local stores and amenities may improve older adults’ health-related well-being (*p* 0.033, 95% CI 0.006, 0.146). Respondents walked to community facilities regularly when the town was easily accessible. It is assumed that the walkability of a neighborhood is influenced by the convenient accessibility to destinations. Seniors who live in walkable neighborhoods generally feel more comfortable. Walking has a plethora of social and health benefits [11,66]. For example, Berke et al. [11] studied neighborhood walkability and its relationship to depressive symptoms and found that the likelihood of developing depression decreased in more walkable neighborhoods, especially among male seniors. Furthermore, regular walking can increase the amount of time that older adults spend exercising, resulting in better physical and mental health [28].

The presence of a neighborhood problem, on the other hand, was significantly inversely related to the health outcomes of the elderly (*p* 0.000, 95% CI −0.299, −0.146). The significant neighborhood problem results indicated that older people who live in a problematic neighborhood were more prone to reporting poor mental health. Previous research evidence suggested that there is an inverse association between health-related well-being and neighborhood problems [15,42]. In this survey, respondents’ reactions to noise, damaged properties, abandoned buildings, graffiti, trash on sidewalks, and illegal activities were measured to assess neighborhood problems. Noise pollution, in particular, has been shown to have a negative influence on older adults’ well-being by interfering with sleep and other important tasks [9,50]. Additionally, the degree of annoyance in the neighborhood, as measured by dilapidated streets and damaged buildings, has been related to an increased risk of developing anxiety and depression [42].

The experience of pedestrian connectivity, such as pedestrian signals, sidewalks, and well-lit neighborhoods, was statistically insignificant in improving older adults’ overall mental health, holding all other variables constant. It appears that respondents in Sunway City do not consider the township to be a pedestrian-friendly living environment. As described in the literature, smooth sidewalks, curbs, pedestrian signals, and accessible sidewalks are some of the pedestrian-friendly design elements that can help improve the physical activity of older adults, as the vast majority of outdoor falls that occur among senior adults are pedestrian and sidewalk related [67,68].

In contrast to the previous findings of Handler [69] and Evans et al. [70] on the relationship between housing conditions and older adults’ mental health status, perceived housing conditions, such as brightness and ventilation inside the dwelling, kitchen and washing area, bathroom, outdoor drainage, and building structure were not significantly related to the mental health status of the elderly. It seems that these housing attributes did not meet the current needs of seniors to support their healthy aging. As mentioned earlier, previous research has found that low brightness in the home increases the risk of accidents and causes the elderly to become disabled and suffer from home injuries [17,71], which in turn causes depression [72]. The bathroom and toilet are also the most dangerous places for a fall [18,71]. Thus, housing should be built to adapt to changing requirements and lifestyles of seniors over time. It is commonly assumed that households select various types of housing depending on their specific needs. For example, they regularly adjust their housing situation as their housing needs or physical conditions change. A senior-friendly dwelling should have features such as a smaller size, step-free access and non-slip flooring in a larger bathroom, and a wide door for wheelchair access [6]. Therefore, there is a need to pay attention to key design variables that contribute to a good home environment for the elderly.

In this survey, elderly women had better mental health than elderly men, in contrast to other researchers’ findings that women were more likely than men to suffer from depression. [46,47,50,63,73]. One explanation could be that most female elderly respondents to this survey were physically active, which had a direct impact on their self-rated mental well-being. It is also worth noting that older adults with an elementary education (*p* 0.000, 95% CI 0.263, 0.685) or a college degree (*p* 0.026, 95% CI 0.019, 0.294) reported higher levels of mental well-being, but not seniors with a secondary education background (*p* 0.102, 95% CI −0.024, 0.272). Being married was significantly related to better mental well-being, consistent with previous findings [48,74] (*p* 0.005, 95% CI 0.047, 0.259). Better mental health was also significantly and positively associated with the ability to drive a car (*p* 0.000, 95% CI 0.993, 1.315). As expected, the greater the number of dependents, the more likely older adults were to have lower levels of mental well-being (*p* 0.003, 95% CI −0.060, −0.012); the literature suggests that the number of owned children in the family is frequently the source of anxiety and stress [75], which may affect mental health. Additionally, homeownership (*p* 0.000, 95% CI −0.602, −0.271) was significantly related to older adults’ mental health status, but the relationship was negative. In general, homeowners have fewer mental health problems [51]; the benefits of homeownership to homeowners has been observed in many housing studies [76]. It would be interesting to learn why the older adults in this study believed otherwise. Respondents in this study were also asked to provide their age, employment status, and racial identification, but no significant differences in age, race, or employment status were reported.

## 4. Conclusions

The benefits to individuals and society are frequently used to justify development policies and strategies to support people to age in place. In the face of age-related health limitations, the value of the built environment in promoting health in old age may increase. As Malaysia’s population ages, it is critical to create an age-friendly environment that helps Malaysian seniors to age healthily. However, only a few studies have examined the impact of perceived home and neighborhood environments on older Malaysians. This research adds to the existing body of knowledge by providing evidence to support the importance of environmental attributes on mental well-being.

According to the findings of this study, perceived accessibility, environmental qualities, and neighborhood problems have a significant effect on the mental health of the elderly. It is important to develop an attractive elders’ neighborhood environment with access to the natural landscape, as this could support older adults’ overall mental health as they age.

There is also a need to create an active living environment that promotes physical and social activities for older adults. The availability of community amenities on a walkable scale promotes active living by optimizing health-related opportunities and involvement, thereby improving the health-related well-being of the elderly. There is a greater understanding that senior people can age successfully if they live in walkable neighborhoods where they can partake in social and physical activities outside of their home. As a result, township planners, architects, and housing developers should design an age-friendly neighborhood with accessible amenities to keep older adults physically and socially engaged as they age.

Although perceived pedestrian connectivity and safety in this neighborhood had negligible effects on mental well-being in this study, the importance of a safe pedestrian network must be considered when designing an inclusive, age-friendly urban environment. Pedestrian connectivity is a critical component of an inclusive urban neighborhood.

Another conclusion from this research is that developers should focus on the development of age-friendly housing projects to address the changing needs and lifestyles of the elderly. Developers also need to design houses that take into account the various life cycle patterns of housing consumption to reflect older adults’ overall life experience rather than building conventional houses.

These findings can serve as a basis for policy decisions and guidelines on resource allocation and neighborhood planning that combine meaningful and supportive neighborhood alternatives for older adults to create a stronger age-friendly community. Such policies and guidelines should also help older adults live independently longer, avoiding or delaying the need for costly long-term care.

## Figures and Tables

**Figure 1 ijerph-19-03595-f001:**
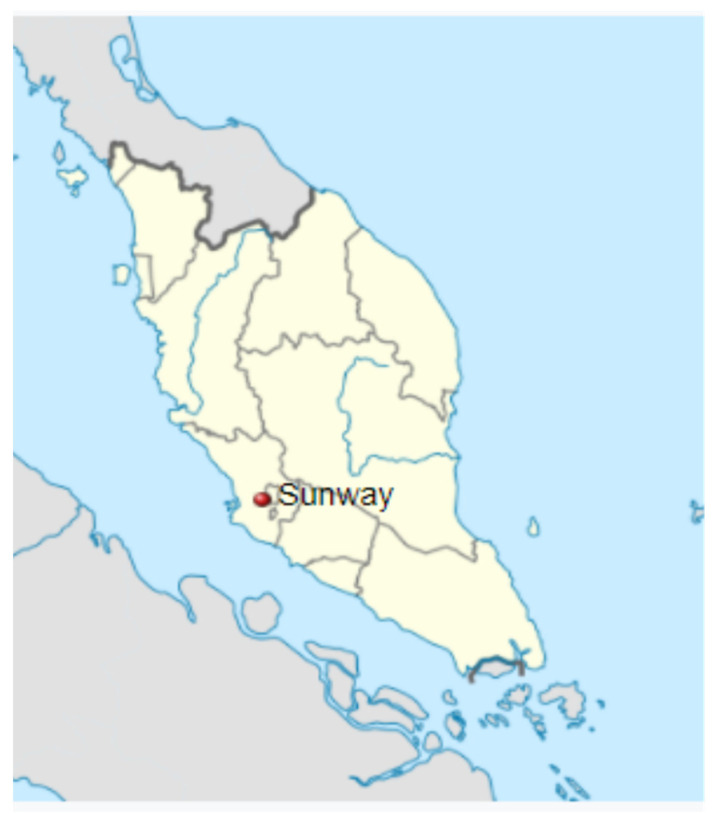
Location of Bandar Sunway. Source: [53].

**Figure 2 ijerph-19-03595-f002:**
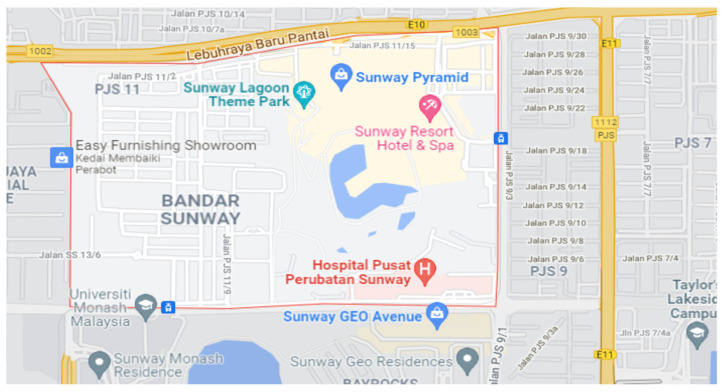
Targeted areas of the study (PJS7, PJS9, PJS 11). Source: [54].

**Table 1 ijerph-19-03595-t001:** Measures of surveyed variables.

	Items	Coding	CR, AVE
Home environment [6,55]	How well are your housing needs met? H1: Kitchen and laundry facilities H2: Ventilation and brightness of the unit H3: Bathroom and toilet H4: Outdoor housing conditions (i.e., building structure, drainage)	1 = poor 2 = average 3 = good 4 = excellent	CR = 0.920 AVE = 0.743
Accessibility [56]	A1: Stores are within easy walking distance of my home A2: Many places are within easy walking distance of my house. A3: It is easy to walk to a transit stop (bus, train) from my home A4: I can do most of my shopping at local stores	1 = strongly disagree 2 = disagree 3 = agree 4 = strongly agree	CR = 0.934 AVE = 0.778
Pedestrian connectivity [56]	PC1: There are sidewalks on most of the streets in my neighborhood PC2: My neighborhood is well lit at night PC3: There are pedestrian crossing signals to help you cross the street. PC4: People can easily see walkers and bikers on the street. PC5: There is a pavement/grass that separates the streets from the sidewalks		CR = 0.924 AVE = 0.709
Neighborhood safety [56]	S1: There is a low crime rate in my neighborhood S2: The low crime rate in my neighborhood makes it safe to go on walks		CR = 0.974 AVE = 0.949
Environmental quality [8]	EQ1: In my neighborhood, there are attractive natural sights. EQ2: There are attractive and interesting buildings in my neighborhood. EQ3: In my neighborhood, there are many attractive natural sights		CR = 0.954 AVE = 0.874
Neighborhood problems [15,57]	NP 1: Noise pollution NP 2: Damage of public or private property NP 3: Abandoned buildings NP 4: Burglary, robbery, violence, and gangsters NP 5: Illegal gambling and drinking in public places	1 = not at all 2 = slightly 3 = moderately 4 = very	CR = 0.958 AVE = 0.792
Mental health [58]	MH1: I was bothered by things that usually don’t bother me MH2: I had trouble keeping my mind on what I was doing MH3: I did not feel like eating; my appetite was poor MH4: I felt depressed MH5: I felt lonely and sad MH6: My sleep was restless	1 = rarely 2 = sometimes 3 = often 4 = most of the time	CR = 0.973 AVE = 0.858

**Table 2 ijerph-19-03595-t002:** Respondents’ demographic information.

Descriptors	Details	Percentage (Frequency)
Gender	Female	50.60 (258)
	Male	49.40 (252)
Marital status	Married	80.20 (409)
	Others (i.e., widowed, divorced, separated, or never married)	19.80 (101)
Retired	No	52.00 (265)
	Yes	48.00 (245)
Tenure status	Own	93.70 (478)
	Rent	6.30 (32)
Driving own car	Yes	83.50 (426)
	No	16.50 (84)
Age	60–70	92.90 (474)
	Above 70	7.10 (36)
Race	Chinese	54.90 (280)
	Malay	33.90 (173)
	Indian	9.40 (48)
	Others (i.e., other indigeneous, non-citizen)	1.80 (9)
Education	Primary	6.70 (34)
	Secondary	54.5 (278)
	College	29.0 (148)
	Others (professional certificates, etc)	9.80 (50)
Household size (average)		4.32 members

**Table 3 ijerph-19-03595-t003:** Results of regression for mental health.

	B	Std. Error	t	*p*-Value	95% LCI	95% UCI
Constant	2.766	0.302	9.152	0.000	2.172	3.360
Home environment	−0.038	0.028	−1.354	0.176	−0.093	0.017
Neighborhood safety	−0.033	0.029	−1.118	0.264	−0.091	0.025
Accessibility	0.076	0.036	2.134	0.033	0.006	0.146
Neighborhood problems	−0.222	0.039	−5.702	0.000	−0.299	−0.146
Pedestrian connectivity	−0.040	0.035	−1.132	0.258	−0.109	0.029
Environmental quality	0.071	0.029	2.447	0.015	0.014	0.129
Retired	−0.049	0.090	−0.541	0.588	−0.227	0.129
Age above 70 years old (ref group)						
Age 60–70 years old	0.098	0.058	1.687	0.092	−0.016	−0.212
Female	0.699	0.042	16.554	0.000	0.616	0.782
Other race (ref group)						
Chinese	−0.203	0.149	−1.366	0.173	−0.495	0.089
Malay	−0.228	0.150	−1.520	0.129	−0.522	−0.067
Indian	−0.155	0.157	−0.988	0.324	−0.464	−0.153
Other qualification (ref group)						
Elementary school	0.474	0.107	4.416	0.000	0.263	0.685
Secondary school	0.124	0.075	1.641	0.102	−0.024	0.272
College	0.156	0.070	2.230	0.026	0.019	0.294
Married	0.153	0.054	2.833	0.005	0.047	0.259
Household size	−0.036	0.012	−2.984	0.003	−0.069	−0.012
Homeowner	−0.437	0.084	−5.179	0.000	−0.602	−0.271
Driving	1.154	0.082	14.062	0.000	0.993	1.315
R Square	0.820					
Adjusted R Square	0.813					
F, *p*-value	117.428, 0.000					

## Data Availability

The data presented in this study are available on request from the author.

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
