# Peer review of "Perceived Environmental Attributes: Their Impact on Older Adults’ Mental Health in Malaysia"

_ijerph, 2022, doi:10.3390/ijerph19063595_

Round 1

Reviewer 1 Report

  • Brief summary

The aim of this paper is to study the impact of internal and external built environment on older adults’ perception of health from a Malaysian township. This quantitative study is based on a stratified sample of 510 persons, aged 60+, who completed a questionnaire. The author is particularly interested in the potential role of “third places”, a concept of place beyond housing and working place. Significant results are found regarding accessibility, neighborhood safety, environmental qualities and neighborhood problems. Cultural and educational-related place and shopping malls are the “third places” with the higher (positive) impact on health perception.

  • General concept comments

This paper addresses a domain (the built environment) interesting the developments of the rising attention of sciences and public policies in different place of the world toward “age-friendliness”. It is focused on a township from a developing country (Malaysia) and is build upon a questionnaire answered by 510 older inhabitants (60+, majority of woman and a very slight majority of whom being retired).  Based on literature, it explores the internal (housing) and external built environment’ influence on the perception of older adult’s health. While some determinants are very strong (neighborhood safety, environmental quality) and are consequently well presented, other have little impact (internal built environment, pedestrian connectivity) and are less discussed. In parallel, the paper explores the types of “third places” (a concept which is not enough defined and discussed; see further) which would have the higher impact on the explanation of the good perception of their own health by older respondents.

Regarding literature, one may be surprised by the use of some references to justify some propositions.

For ex., p.2, note 11 referring to the work of Handler has little to say with the compared satisfaction of older adults between external and internal built environment.

The same page starts by referring to “many recent students…”. Beyond the fact that “students” should probably be “studies”, there is only 1 ref. to support this proposition.  The third paragraph of the page mentions “improving the external built environment is more difficult than improving the internal built environment” should be sourced; we might even consider this is a personal (or political) opinion, but not a scientific consideration; it might eventually make sense, but after sources and probably more in a conclusion section.

Still on page 2, “The practice of making the built environment as accessible to as many people as possible is known as accessibility”. This is not correct. Such practice refers to “universal accessibility” (if you would have completely read the Handler’s book, you would have understood that such “universality” is, according to her, impossible). Furthermore, you might mention the non-built dimension of accessibility too, even it is not a central issue of your paper. Why ? Because many difficulties in projects and aims like the one supported in the paper (measuring the impact of X features of built environment on Y aspect of older adult’s life) has to do with social representations. World Health Organization consequently brings the attention in its World Report of 2015 on the “ageism” aspect related to it.

More generally regarding the example of practices of “age-friendliness” or “age-friendly features” like bathrooms, the counter-argument also exist: the more “age-specific” a feature you make, the more ageism it might implicitly/explicitly produced (https://www.researchgate.net/profile/Todd-Morrison-2/publication/7687833_Stereotypes_of_Ageing_Messages_Promoted_by_Age-Specific_Paper_Birthday_Cards_Available_in_Canada/links/02e7e52e343b6805c8000000/Stereotypes-of-Ageing-Messages-Promoted-by-Age-Specific-Paper-Birthday-Cards-Available-in-Canada.pdf ; more generally, this refers to the “activation of the ageing stereotype” : Voss, P., Bodner, E., & Rothermund, K. (2018). Ageism: The Relationship between Age Stereotypes and Age Discrimination. In L. Ayalon & C. Tesch-Römer (Eds.), Contemporary Perspectives on Ageism (pp. 11–31). Springer International Publishing. https://doi.org/10.1007/978-3-319-73820-8_2).

Last but not least, the concept of “third place” is not conceptually presented and discussed. This concept has been forged in the US by Oldenberg theory “The great good place” in 1989. The paper only offers a large definition. but what is your def ? And it is about social life value

For ex., and because the paper explicitly refers itself to the geography of a developing country, one might for ex. question the universality of such concept. Another question turns around the author’s own translation of the concept. Last but not least, the original concept insists on the “social life value”, next to the centrality of family and work aspects of life. If the paper interestingly shows the higher impact of “Cultural and educational-related place” and of “shopping malls” on health perception, we might wait more discussion between the similarities and differences of such places. The author starts the discussion; however, it might be more deeply developed (because a “shopping mall” is not exactly the same as a “cultural place”; in ref. to Bourdieu’s theory, one may also consider that not all kinds of population go to such “cultural places”, etc.).

Regarding hypothesis, the author seems to hesitate between focusing on the built aspect of environment and/or on the types of “third places”.

It is also a bit unclear to my understanding (and this might really be my own qualitative bias) how far the study really discusses the “internal built environment” aspect, which is referring to the housing features. My doubt might probably be link with the little effect of such variable in the three models (p.6). Which is not commented, if I am correct? To formulate it clearly: I wonder if the paper is not too ambitious is willing to grasp and analyze, simultaneously, the (private, inner) “internal built env.” And the (public, open) “external built env.” The literature is generally divided, even if the “ageing in place” notion explore the links between connection of private/public places.

This might also be part of a first publication by the author  : https://www.researchgate.net/publication/357715862_Residential_environment_third_places_and_well-being_in_Malaysian_older_adults#fullTextFileContent  (Tan, T. H., & Lee, J. H. (2022). Residential environment, third places and well-being in Malaysian older adults. Social Indicators Research. https://doi.org/10.1007/s11205-021-02856-8)

One might also wait more information regarding the potential originality of this example from a developing country (because the paper explicitly refers to scarcity of work on “age-friendliness” in such areas (which I totally agree and therefore suggest to explore).

Regarding results. First, I do not see the average age of the sample. However, we notice that older adults appear to be relatively young, no? What is the impact of the age? Could we generalize the result to “the elderly” as expressed in conclusion?

Second, what is the effect of gender? And the effect of religion of respondent? In a multicultural country like Malaysia, such potential aspect might also be of interest (for ex. the access to public space is not similar for men/women in Mueslin; another clue of this refers to the results regarding the type of classes in Educational space, like QiGong and TaiChi, which might also support cultural specificities).

Third, the third model refers to the influence of sociodemographic characteristics. This model explaining 71,4% of the perception of health by older adults. First, do you have any suggestion regarding the missing 28,6%?  Second, regarding sociodemographic characteristic, as a French sociologist, I am a bit surprised you do not have the information regarding the professions (or the educational level), as it might refer to a social class analysis. Indeed, one of the biggest issues of the paper is the potential invisible bias. This is also visible in the literature presentation. Shortly said: to consider one’s health as good, it is better to live in a safer environment without “neighborhood problems”. Or to live in a place with parks and trees.

If one follows this observation, the conclusion of the paper might be to support “gated community” where people consider themselves as more secure, their environment safer and of higher quality (Age, Meaning, and Place: Cultural Narratives and Retirement Communities. Stephen Katz and Kevin E. McHugh. A Guide to Humanistic Studies in Aging, eds. Thomas R. Cole, Ruth Ray and Robert Kastenbaum. Johns Hopkins University Press, 2010). Don’t you think there is resource/socioeconomic biais in the answer, the richer giving the “better” answers?

Last but not least and essential: in comparison with your just published paper (Tan, T. H., & Lee, J. H. (2022). Residential environment, third places and well-being in Malaysian older adults. Social Indicators Research. https://doi.org/10.1007/s11205-021-02856-8), what is original in this paper, except the larger size of the sample? And the change of focus from “well-being” to “health”, as these two dimensions might be referred to the same indicators (however, we can’t decide it as I couldn’t access the full version of the first paper.) Reading your abstract of the already published paper make me believe nothing is really new here.

  • Specific comments

P1, line 32 : “concept of an age-friendly built environment” ; it ignores the ageism/non-built aspect. I would suggest to delete “built” here.

P1 l 34 :  “Annual Global Retirement Index”. What is it ?

P2 l 47 : “students” or “studies”?

P2 l 49 : Building Design Guidelines : missing in ref.

P3 l 131 : first occurrence of “third place” in the text. Refer to Oldenberg 1989 ? Discuss it slightly?

P6 l 212/ In the model, “b4Ci” shouldn’t it be “b4Si” ?

Author Response

Attached, please find the responses to Reviewer 1. Thanks again for your valuable comments and feedback.  

Reviewer 2 Report

Review

“Built Environment Determinants: their Impact on Older Adults' Overall Health in an Age-Friendly Township in Malaysia”

I agree with the background of this study to understand the age-friendly urban environment for health promotion amid the growing elderly populations. Many studies have struggled with these topics, and more research is still needed. In order to strengthen academic contributions of this unique condition of the study site, the overall research design needs to be improved. The current state of this manuscript has several limitations in terms of selecting variables, setting the statistical hypothesis, and interpreting the results of regression models.

  1. Abstract does not deliver core information and findings.

(Line 13) 'A questionnaire’ which gives raw data for this study needs more detailed descriptions.

(Line 13) The confidence level of the determinants, mentioned as 'accessibility, neighborhood safety, environmental qualities and neighborhood problems’ needs to be explained and properly informed. Please add the p-value of each variable.

(Line 15) There are quite disconnected sentences about this study’s critical determinants and third places. This problem might be related to the variable selection.

  1. Introduction section needs to be re-structured in order to clarify the theoretical framework of the study.

(Line 48-51) needs to cite the documents or references, ‘Building Design Guidelines for Malaysian Elderly’.

(Line 64-72) This paragraph suggested ‘bathroom flooring’ as an example of design element of internal built environment. However, surveyed answer was just categorized into 4-point Likert scale. The readers might have a misunderstanding that your survey asked a lot of small elements in the internal built environment. I think this paragraph needs to explain the overall perceived condition of internal built environment, or suggests proper examples.

(Line 73-130) Each variable in this study tends to be explained at each paragraph. It is good to understand. However, there are other numerous determinants in addition to those variables you suggested. The author needs to explain how those variables were selected and why others were excluded. These kinds of study, which focus on finding significant determinants, should suggest the logic of variable selection. Theoretical framework might be a key to explain how preceding literatures addressed the overall association/relationship of determinants. In addition, the idea of variable structure from other references should be written in Introduction section. If you conducted the variable selection for this study, it should be explained in Methods section.

(Line 91) Pedestrian connections --> Pedestrian connectivity

  1. Study area

More information of the study area, Sunway City, Malaysia, need to be provided for the global readers who are not familiar with it. For example, it would be helpful to draw the figure, the map of Sunway City: the distribution of senior-friendly houses; three districts named PJS 7, 9, 11; the locations of third place; the pedestrian and transit network.

(Line 148-150) needs a citation about the issue which was addressed by the urban planner.

  1. Special/generalized interpretations from the findings of this case study

How would you locate the research findings of this site, i.e., are there special or general characteristics in the research findings compared to those in other cities in Malaysia? Or, could you explain which findings in this site would newly inform foreign readers and global academia?

  1. Questionnaire

(Line 147) How many older adults participated in the survey? Please add: Older adults (N = 510)

Is the questionnaire designed only for this study? or multiple purposes?

Could you submit the original questions at the appendix? If it cannot be added in the appendix, at least it needs to be explained regarding the purposes, subjects, interviewers, # of questions, structured or unstructured, etc of the questionnaire.

In addition, the process how you coded the answers to categorize into the items in each variable needs to be provided in the manuscript.

  1. Coding items of each variables

The manuscript does not give any detail for the questionnaire. (Table 2) It is insufficient to understand the original questions for each variables. The manuscript needs to address not only citing the references Tan[26], Cerin, et al.[27], but more explaining how those answers had been coded.

(Line 175) 'selected with slight modifications from previous studies’, it also needs more accurate explanation about the ‘modifications’. This is directly related to the main variables in the regression models.

  1. Selecting variables

The determinants are categorized in four parts: internal building, neighborhood environments, type of third place, socio-demographic characteristics of subjects. Considering other variables, the study should address the variable selection process. Please consider several questions: why aforementioned determinants were selected; why others were excluded; why the multiple elements of internal building were compiled into a single variables.

This study compares those variables at the same level in Model 3. However, it is too risky to compare those variables as dependent variables. It is better to compare between the similar variables, for example, the cultural third place and the worship third place could be compared.

Internal built environment (building) is coded as a single determinant, but I think it is not clear. Every different building has different status of built environment. If the building is the house of each subjects, it should be written in the manuscript. Even though it meant a single building, this answer might imply multiple aspects or elements of internal built environment. I recommend to reconsider that a lot of subjects’ answers resulted in a single determinants.

Each determinants also have multilateral effects for the perceived health status. For example, bathroom flooring, which is mentioned in Introduction, would cause the physical damage by slippery, but it is less relevant with mental health.

  1. Explaining the results

(Line 237) According to the comment “after controlling for types of third places and sociodemographic characteristics”, did this study differentiate the determinants as control variables and dependent variables?

(Line 234 -240) Comparing model 1 and model 3, the author noted the different odds of being in good health. However, it is not precise because there might be the multi-colinearity between the variables. For example, it is possible that the answers are similar for the determinant ‘accessibility’ and ‘pedestrian connectivity’. If there is a relationship between variables, the value of Exp(B) of variables were effected by each others. Highly recommend to compare the p-value and find significant variables.

  1. Regression model and Hypothesis

Please reconsider the structure of regression model and research hypothesis.

With careful variable selection and result interpretation, this study might suggest which variables are significant for perceived health status. Then those findings could show the significant level of determinants.

The limitation of this study cannot explain the effects of external variables. If there were much significant variables which were not included in the regression model, the results were distracted from the actual correlation for perceived health status. I assume that other variables, i.e. genetic factors, current illness, house building types, need to be tested for the correlation.  

Using regression analysis to find the influence of variables is the correct approach, but it is recommended to interpret carefully when designing the three regression models and their variables.

Author Response

Attached, please find the responses to Reviewer 2. Thanks again for your kind and valuable comments and feedback. 
